# Diagnostic Clues for Women with Acute Surgical Abdomen Associated with Ruptured Endometrioma

**DOI:** 10.3390/jpm13081226

**Published:** 2023-08-02

**Authors:** Jihyun Keum, Won Moo Lee, Joong Sub Choi, Jaeman Bae, Seongsil Cho, Bo Kyeong Kang

**Affiliations:** 1Division of Gynecologic Oncology and Gynecologic Minimally Invasive Surgery, Department of Obstetrics and Gynecology, Hanyang University College of Medicine, Seoul 04763, Republic of Korea; goldkjh@hanyang.ac.kr (J.K.); choiyjjy1@hanyang.ac.kr (J.S.C.); obgybae@hanyang.ac.kr (J.B.); ssv1226@gmail.com (S.C.); 2Department of Radiology, Hanyang University College of Medicine, Seoul 04763, Republic of Korea; msbbogri@hanyang.ac.kr

**Keywords:** endometrioma, ruptured state, CA 125, CA 19-9, CRP

## Abstract

(1) Background: An investigation of the preoperative diagnostic clues used to identify ruptured endometrioma by comparing the ruptured and unruptured states in patients who underwent laparoscopic operations due to endometrioma. (2) Methods: Patients with ruptured endometriomas (14 patients) and unruptured endometriomas (60 patients) were included, and clinical symptoms, laboratory findings, and radiological findings were analyzed. (3) Results: There were no significant differences in age, parity, last menstrual cycle days, or median size of endometrioma between two groups (group A: ruptured; group B: unruptured). The median serum level of CA 125 was 345.1 U/mL in group A and 49.8 U/mL in group B (*p* = 0.000). The median serum levels of CA 19-9 in group A and B were 46.0 U/mL and 19.1 U/mL, respectively (*p* = 0.005). The median serum level of CRP in group A was 1.2 g/dL, whereas it was 0.3 in group B (*p* = 0.000). ROC analysis showed that the optimal CA 125 cutoff value was 100.9 U/mL; the optimal CA 19-9 cutoff value was 27.7 U/mL; and the optimal CRP cutoff value was 1.0 g/dL. (4) Conclusions: Ruptured endometrioma can be diagnosed preoperatively using a combination of clinical symptoms, laboratory findings, and radiological findings. If a physician suspects a ruptured endometrioma, surgery should be performed to ensure optimal prognosis.

## 1. Introduction

Endometriosis is a condition characterized by the presence of small growths that are frequently found on pelvic structures, such as the ovaries, bladder, and cul-de-sac (CDS). In certain instances, these growths can progress and form larger cystic lesions referred to as endometriomas, primarily within the ovaries. A considerable number of individuals with endometriosis seek medical assistance as they experience symptoms such as chronic pelvic pain, dysmenorrhea, dyspareunia, dyschezia, and infertility [1]. As a result, many of these patients undergo planned surgical interventions to address their condition [2].

However, in cases where an endometrioma ruptures, the release of its contents can lead to various symptoms, including the sudden onset of abdominal pain, abdominal distension, nausea, vomiting, and an aggravation of peritoneal irritation over time. This acute abdominal pain is primarily caused by chemical inflammation, resulting from the peritoneal irritation caused by the presence of old blood from the ruptured endometrioma. If these symptoms are present, a blood test can indicate an elevation in the white blood cell count, as well as increased levels of CA 125 and CA 19-9 tumor markers [3,4].

Computed tomography (CT) scans reveal the presence of ovarian tumors accompanied by abnormal fluid accumulation in the abdominal cavity. Distinguishing between advanced epithelial ovarian cancer (EOC), hemoperitoneum, pelvic tuberculosis, and tubo-ovarian abscess poses a challenge due to the aforementioned imaging findings, as well as the observed elevation in CA 125 serum levels among patients [5].

While delaying surgical treatment can potentially result in the development of intra-abdominal adhesions and other complications, it is worth noting that the diagnosis of this condition is occasionally prone to misdiagnosis. Furthermore, due to the inherent risk of advanced epithelial ovarian cancer, a considerable number of physicians lean towards postponing surgery rather than opting for immediate or emergency surgical intervention [5,6].

To the best of our knowledge, there have been no prior studies conducted to specifically investigate potential diagnostic indicators for identifying ruptured endometrioma preoperatively. Therefore, the main objective of this study was to examine and compare certain factors among women with ruptured endometriomas and those with unruptured endometriomas, with the aim of identifying preoperative diagnostic clues that could potentially predict the presence of a rupture endometrioma.

## 2. Materials and Methods

### 2.1. Patient Selection and Data Collection

We conducted a retrospective study of 74 patients who were diagnosed with endometrioma between January 2013 and December 2014. The patients were divided into two groups: Group A consisted of individuals with ruptured endometrioma, while group B included those with unruptured endometrioma. The study specifically included patients who underwent laparoscopic procedures, such as ovarian cystectomy or unilateral salpingo-oophorectomy, while excluding those who underwent hysterectomy with or without bilateral salpingo-oophorectomy due to the presence of coexisting benign gynecological conditions. The radiological review of the cases was conducted by an expert radiologist (B. Kang).

In this study, we conducted an analysis of medical records encompassing various aspects, including demographic characteristics, such as age, parity, body mass index, and the menstrual cycle day from the last menstrual period at the time of surgery. Additionally, clinical and laboratory data were examined, which involved documenting the initial symptoms reported by the patients, as well as measuring the serum levels of cancer antigen 125 (CA125), carbohydrate antigen (CA19-9), and C-reactive protein (CRP). Furthermore, surgical outcomes, revised American Society for Reproductive Medicine (ASRM) scores, gynecologic sonography findings, and CT scans were also evaluated for both groups. This study was approved by the Institutional Review Board of Hanyang University Hospital.

### 2.2. Statistical Analysis

The Mann–Whitney U test was utilized to evaluate the differences between continuous variables, including CA125, CA19-9, CRP, BMI, and the size of the mass. Fisher’s exact test was employed to assess categorical variables, such as age, parity, and menstrual cycle day. To determine the optical cutoff values for diagnosis, a receiver operating characteristic (ROC) curve was generated for the serum levels of CA 125, CA 19-9, and CRP. The sensitivity, specificity, and accuracy of each cutoff value were evaluated. Statistical significance was defined as *p* < 0.05. For all statistical analyses, SPSS 13.0 for Windows (SPSS Inc., Chicago, IL, USA) and MedCalc version 15 (MedCalc Software BVBA, Acacialaan, Ostend, Belgium) were used.

## 3. Results

### 3.1. Clinical Characteristics

Among the total of 74 patients included in this study, 14 individuals were classified into group A, while the remaining 60 patients were classified into group B. All 14 patients in group A experienced an acute onset of lower abdominal pain (LAP) and sought immediate medical attention in the emergency department. In contrast, among the patients in the unruptured group (Group B), only eight individuals (13.3%) presented to the emergency room with lower abdominal pain. The majority of group B patients, comprising a total of 52 cases (86.7%), sought care in an outpatient clinic. Among those who visited the outpatient clinic, 14 cases (23.3%) reported lower abdominal pain, while 13 cases (21.7%) experienced dysmenorrhea. Notably, the most frequent presentation was the incidental discovery of endometrioma in asymptomatic patients, accounting for a total of 25 cases (41.7%).

In group A, the median serum level of CA 125 was 345.1 U/mL (range, 70.8–3363), while in group B, it was 49.8 U/mL (range, 9.2–317.4). This difference was found to be statistically significant (*p* = 0.000). Similarly, the median serum level of CA 19-9 in group A was 46.0 U/mL (range, 1.0–1158) compared to 19.1 U/mL (range, 0.6–176.2) in group B. Again, this disparity was also deemed statistically significant (*p* = 0.005). The serum level of CRP in group A was 1.2 g/dL (range, 0.3–16.6), whereas it was 0.3 g/dL (range, 0.3~12.0) in group B, and this difference was also found to be significant (*p* = 0.000). Regarding the median BMI values, group A had a median of 19.4 kg/m^2^ (range, 17–23.4), while group B had a median of 21.5 kg/m^2^ (range, 17–32). Although these values were significantly different (*p* = 0.048), it is important to note that both values fell within the normal range of body weight (Table 1).

### 3.2. The Difference of Surgical and Radiologic Findings between Two Groups

There were no significant differences observed in the locations of the masses between group A and group B. The median size of the masses was 6.5 cm (range, 5.0–8.0) in group A and 6.0 cm (range, 3.0–10.0) in group B, and this difference was not found to be statistically significant (*p* = 0.277). In group A, laparoscopic cystectomy was performed on all patients, while in group B, 56 patients (93.3%) underwent laparoscopic cystectomy and only four patients (6.7%) underwent unilateral salpingo-oophorectomy. There were no significant differences observed in operative procedures between the two groups (*p* = 0.424). The median ASRM score was 36 (range, 12–84) for group A, and it was also 36 (range, 10–140) for group B. Abdominal CT scans were conducted for all 14 patients in group A. However, only two patients were diagnosed with ruptured endometrioma preoperatively (14.3%). The CT scans revealed that all patients in group A had an adnexal mass with an irregular surface and fluid echo in the cul-de-sac (CDS). Additionally, peritoneal infiltration was observed in all patients. Among the patients, eight showed fluid echo in the right paracolic gutter, six exhibited fluids in the left paracolic gutter, three had it in the subhepatic space, and one patient displayed it in the left subphrenic area. Fluid loculation was observed in eight out of the fourteen patients (Table 2).

### 3.3. The Optimal Cutoff Value

The ROC analysis showed that the optimal cutoff value for CA 125 was determined to be 100.9 U/mL. As this cutoff, the sensitivity was found to be 85.7%, the specificity was 83.3%, and the area under curve (AUC) was 0.92 (95% CI, 0.834–0.970, *p* < 0.0001). Similarly, the optimal cutoff value for CA 19-9 was identified to be 27.7 U/mL, resulting in a sensitivity of 78.6% and a specificity of 67.9%. The AUC for CA 19-9 was calculated as 0.745 (95% CI, 0.627–0.842, *p* = 0.003). Furthermore, the optimal cutoff value for CRP was determined to be 1.0 g/dL, with a sensitivity of 64.3%, specificity of 96.6%, and an AUC of 0.79 (95% CI, 0.676–0.876, *p* = 0.0004) (Figure 1).

## 4. Discussion

In this study, we have observed disparities in clinical manifestations and radiographic observations, along with notable disparities in serum concentrations of CA 125, CA 19-9, and CRP between patients diagnosed with ruptured endometriomas and those diagnosed with unruptured endometriomas.

Endometriosis is characterized by the existence of endometrial-like tissue beyond the confines of the uterus, associated with a chronic and inflammatory response. The pathogenesis of endometriosis remains a topic of debate, but mounting evidence suggests a substantial involvement of immunological and inflammatory factors in the onset and progression of endometriosis [7].

CA125 is a mucinous glycoprotein with a high molecular weight, derived from coelomic epithelia found in various locations, such as the endometrium, fallopian tube, ovary, and peritoneum [8,9]. Tests evaluated for screening have not detected ovarian cancer at an early enough stage to reduce mortality and have led to unnecessary surgical procedures for false-positive results. Studies show that CA125 may predict ovarian cancer, but its usefulness is hampered by its limited specificity and very low positive predictive value. Furthermore, it is important to note that elevated levels of CA125 can be observed not only in cases of ovarian cancer but also due to several benign causes, and abnormal levels can also be attributed to non-gynecologic factors. Among the gynecological causes that can lead to increased CA125 levels are endometriosis, adenomyosis, pelvic inflammatory disease, and ovarian hyperstimulation syndrome. Furthermore, non-gynecological causes encompass a wide range of conditions, including multi-visceral tuberculosis, liver cirrhosis, tuberculosis peritonitis, renal failure, pancreatitis, and others [10,11]. It is worth mentioning that elevated CA125 levels have been associated with various malignant disorders other than ovarian cancer, such as uterine cancer, fallopian tube cancer, rectal or bladder cancer, pancreatic carcinoma, breast cancer with peritoneal metastasis, and advanced hepatocellular carcinoma [11].

As mentioned previously, CA125 levels can be elevated due to various causes, but its primary applications in the field of gynecology are primarily focused on ovarian cancer and endometriosis. However, it is important to note that the use of CA125 for the diagnosis of endometriosis is not currently recommended [12]. This recommendation stems from the fact that CA125 exhibits low sensitivity (52%, 95% CI 38 to 66%) in accurately diagnosing endometriosis [13]. These findings have been consistently supported by systemic reviews and meta-analyses, which have reported similar results [14].

Despite its limitations in diagnosing endometriosis, CA125 can still be utilized to assess the effectiveness of treatment or identify the recurrence of endometriosis [13,15]. Furthermore, a recent study demonstrated a significant association between serum CA125 levels and the severity of endometriosis, revealing that women with deep infiltrating endometriosis exhibited substantially higher serum CA125 levels compared to those with superficial lesions [16].

In the case of patients with endometriomas, the highest recorded serum CA125 value in a patient with an unruptured endometrioma was 7900 IU/mL, while it reached 9391 IU/mL in a patient with ruptured endometrioma [10,17]. A recent study revealed that approximately half of all patients with ruptured endometrioma exhibited abnormal CA 125 values, defined as levels equal to or exceeding 35 IU/mL [18]. In comparison, patients with unruptured endometriomas had CA125 values ranging from 2 to 1845 IU/mL [18].

Nevertheless, when comparing the mean values between the two groups, we observed that it was 40.0 IU/mL for the ruptured group and 29.0 IU/mL for the unruptured group; however, this disparity did not reach statistical significance. In another study, only 51% of patients with ruptured endometrioma displayed elevated serum CA 125 levels, with a mean value of 278.48 IU/mL (range: 85.2–2844.3) [19,20]. In contrast to the findings of previous studies, our investigation revealed that all patients in the ruptured group exhibited abnormal CA125 levels, ranging from 70.8 to 3363 IU/mL. The median value of CA125 in the ruptured group was 345.1 IU/mL, while it was 49.8 IU/mL in the unruptured group. Importantly, the difference between these two groups was statistically significant (*p* = 0.000).

These findings are likely to be related to the CA 125 concentration in the cystic fluid of endometriomas [21]. Although the CA 125 concentration in cystic fluid is very high in endometriomas, peripheral absorption is prevented by the cystic wall [21]. However, the rapid increase in the CA 125 level in the serum after cyst rupture may be due to the release of CA 125 in the cystic fluid onto the peritoneal surface [22].

Another possible explanation for this observation is that the serum CA 125 level is associated with the size of the tumor. A previous study showed a positive correlation between the serum CA 125 level and the maximal tumor diameter of the tumor, with a correlation coefficient of 0.319 and a *p* = value of less than 0.0001 [18]. However, in our study, we did not find a significant difference in tumor size between the two groups. Nonetheless, it is worth noting that intraabdominal fluid was observed in all patients with ruptured endometrioma. These results suggest that the cyst size prior to rupture may be larger in this group compared to the unruptured group.

CA19-9 is a cell surface glycoprotein complex that is primarily associated with pancreatic ductal adenocarcinoma [23]. CA 19-9 is associated with various diseases similar to CA 125. It is normally synthesized by human pancreatic and biliary ductal cells as well as gastric, colon, uterus, endometrial and saliva epithelia cells [24]. CA 19-9 is excessively produced in a broad range of both benign and malignant conditions, including pancreatic cysts, diabetes mellitus, liver fibrosis, benign cholestatic disease, and other urological, pulmonary and gynecological disease [23]. Elevated level of CA 19-9 has also been observed in benign and malignant gynecologic diseases, especially in mature cystic teratoma and mucinous ovarian tumors [25,26,27,28]. There are several pieces of literature discussing the relationship between CA 19-9 and endometriosis. One study suggests that CA 19-9 can be helpful when used alongside CA 125 for the purpose of diagnosing endometriosis and monitoring its treatment [29]. Another study reports that the combined measurement of CA 19-9 and HE4 may be associated with endometriosis-related ovarian cancer [30]. Serum CA 19-9 has been shown to be elevated according to clinical stage in patients with endometriosis [31,32]. One prospective study showed that the mean CA19-9 level increased throughout all stages of endometriosis, whereas the mean level of CA125 only increased in ASRM stages II, III, and IV [31]. The serum levels of CA 19-9 and CA 125 have been shown to be significantly higher in patients with stage III and stage IV disease compared to patients with stage I disease, and CA 19-9 has been proposed to be a valuable marker to predict severe endometriosis when used with CA 125 [31]. However, other previous studies have reported that CA 19-9 had low sensitivity (8–34% vs. 27–49%, respectively) and high specificity (94–100% vs. 92.3–97%, respectively) compared to CA 125, which reduces the predictive value of CA 19-9 in endometriosis [25,32,33].

In our present study, while the serum CA 19-9 level appeared to be within the normal range in the unruptured group, we observed a statistical difference in the median CA19-9 level between the ruptured and unruptured groups (*p* = 0.005). Moreover, we identified that the optimal cutoff value for CA19-9 in diagnosing ruptured endometrioma was determined to be 27.7 IU/mL, with a sensitivity of 78.6% and specificity 67.9%. These findings indicate a higher sensitivity and lower specificity of serum CA19-9 level for the diagnosis of ruptured endometrioma compared to previous reports, underscoring it utility as a valuable biomarker. It is worth noting that this disparity in diagnostic performance might be attributed to the specific patient population included in our study. Unlike previous studies, our analysis focused solely on patients with endometrioma, excluding cases with concomitant leiomyoma, adenomyosis, or other benign ovarian cysts. This more selective patient inclusion criteria may have contributed to the enhanced sensitivity observed in our results. However, it is important to note that our study specifically focused on patients with either ruptured or unruptured endometrioma, intentionally excluding individuals who had undergone procedures, such as myomectomy, total hysterectomy, bilateral salpingo-oophorectomy, or other ovarian surgeries related to combined benign gynecologic conditions. This deliberate patient selection was aimed at minimizing potential confounding factors from other markers and optimizing the diagnostic accuracy of CA19-9 in the context of endometrioma.

Several studies have provided evidence supporting the association between endometriosis and an inflammatory response, which leads to the upregulation of various inflammatory-related factors [34,35,36]. Endometriosis is connected to the activation of specific components of the immune system, particularly macrophages, along with the secretion of cytokines, the promotion of angiogenic factor formation, and the involvement of specific types of white blood cells, namely T cells and B cells [37]. CRP (C-reactive protein) serves as a valuable biomarker in the identification of inflammation [38]. Given the inflammatory nature of endometriosis, several studies have delved into exploring the potential connection between CRP and this condition. In a recent prospective study, the serum levels of CRP were compared between an endometriosis group and a control group, but no statically significant difference was observed between the two groups [39]. Subgroup analysis specifically comparing superficial lesions and deep infiltrating endometriosis failed to demonstrate any significant differences [39]. Additionally, in another study, as well as in a case–control study encompassing 350 endometriosis patients with endometriosis and 694 control subjects, no associations were identified between endometriosis and plasma interleukin-1, soluble tumor necrosis factor alpha receptor, or CRP levels [40]. Of particular interest, a study conducted to investigate the variations in CRP, CA125, and hemoglobin levels in women with endometrioma and endometriotic nodules revealed compelling results. The findings of this study demonstrated a notable disparity between the two groups. Specifically, women diagnosed with deep infiltrating endometriosis exhibited significantly elevated levels of both CRP and CA 125, whereas their hemoglobin levels were found to be lower [41]. Numerous studies have been conducted to explore the correlation between endometriosis and CRP; however, this particular study stands out as the first to specifically investigate ruptured endometriosis. In our study, we observed that the serum CRP level in the ruptured group was 1.2 g/dL (range 0.3–16.6), whereas it was 0.3 g/dL (0.3–12.0) in the unruptured group; this difference was statistically significant (*p* = 0.000). Furthermore, through the use of ROC analysis, we determined that the optimal threshold for CRP was 1.0, resulting in a sensitivity of 64.3% and a specificity of 96.6%. These compelling findings strongly indicate that CRP can serve as a valuable marker for the diagnosis of ruptured endometrioma.

Abdominal CT scans are typically employed as the primary imaging modality when women present to the emergency room with acute abdominal pain, aiming to facilitate the initial evaluation. However, diagnosing ruptured endometrioma can be challenging due to the nonspecific nature of CT findings [42]. In a prior study, it was reported that ruptured endometriomas exhibited an average diameter of 7 cm and displayed a multilocular structure with thicker walls in comparison to functional cysts [43].

Ancillary findings observed on CT scans have indicated the presence of fluid, loculated ascites, and peritoneal infiltration primarily confined to the pelvic cavity [43]. However, it is important to note that these findings can also be observed in cases of epithelial ovarian cancer, peritoneal tuberculosis, and ruptured endometrioma [5]. Furthermore, when the serum CA125 level surpasses 300 IU/mL, there is a risk of misdiagnosing an endometriotic cyst as an EOC [44]. In our present study, we conducted a comparison between the sizes and shapes of ruptured and unruptured endometriomas, and our findings aligned with those reported in previous studies. However, we identified a slight variation in fluid collection patterns. Specifically, we observed that all patients in our study exhibited intra-abdominal fluid within the pelvic cavity, and notably, 50% of these individuals (seven out of fourteen) demonstrated fluid extension into the paracolic gutter or subphrenic area. In addition, loculated fluid collection was observed in 57% (8/14) of the patients, and peritoneal infiltration was observed in all patients. These findings show that endometriotic cystic fluid is closely related to inflammation [34,35,36]; furthermore, severe peritoneal irritation may lead to the formation of ascites, and the accompanying inflammation can worsen fluid loculation or peritoneal infiltration.

If a ruptured endometrioma is left untreated or if there is a delay in diagnosis, the leaked endometriotic fluid can lead to the obliteration of the cul-de-sac (CDS) and the formation of peritubal and periovarian adhesions. These adhesions can subsequently result in chronic pelvic pain and subfertility [45]. Therefore, it is crucial to ensure early detection and prompt treatment to prevent these complications from arising. Notably, a significant disparity in future fertility has been reported between patients who underwent surgery within 72 h of rupture and those who did not, with a fertility rate of 35% compared to 0%, respectively (*p* = 0.005) [19]. Furthermore, a study with long-term follow-up has demonstrated that emergency surgical intervention is necessary to improve the prognosis in patients with a ruptured endometrioma [20].

Various risk factors contribute to the recurrence of endometriosis, including the severity and type of endometriosis, the surgical procedure employed, the skill of the surgeon, the specific hospital where the surgery was conducted, and the postoperative treatment provided [45]. However, the presence of residual endometriotic lesions is considered the most significant factor [46,47]. In the case of ruptured endometrioma, inflammation, and organ adhesion tend to worsen over time, compromising the surgical completeness and increasing the likelihood of recurrence. Therefore, it is crucial to prioritize early detection and treatment of ruptured endometrioma to enhance the prognosis and reduce the risk of recurrence.

Additionally, it is important to consider the possibility of malignancy when an ovarian tumor is present and there is an elevation in serum tumor markers, such as CA 125 or CA 19-9. Gynecologic causes of hemoperitoneum include ectopic pregnancy, hemorrhagic corpus luteal cyst, endometriosis, and uterine or ovarian neoplasms [48]. However, acute abdominal symptoms due to ovarian malignancy is rare [49]. Among these causes, germ cell tumors and sex cord stromal tumors are the most common ovarian malignancies associated with spontaneous hemoperitoneum [50,51]. Germ cell tumors are typically associated with serum tumor markers, such as human chorionic gonadotropin(HCG) and alpha-fetoprotein (AFP) [52]. Sex cord-stromal tumors utilize various hormones, such as estradiol, inhibin, testosterone, and androstenedione, as tumor markers [50]. In cases of epithelial ovarian cancer, which is the most common malignancy associated with elevated CA 125 levels, it may be challenging to distinguish it from a rupture of an endometrioma. However, there are several important clues to differentiate between them. Firstly, epithelial ovarian cancer is rare in individuals under 40 years of age, with over 90% of cases occurring in those over 40 years old. Additionally, the incidence of epithelial ovarian cancer also increased with age, peaking at around 70 [53]. This study, being retrospective, only included women with confirmed biopsies of endometriomas, and the average age of the ruptured and unruptured groups was 30.575 years and 33.83 years, respectively, which is younger than the typical onset age for epithelial ovarian cancer. Furthermore, clear cell ovarian cancer, which is a rare histologic subtype, primarily develops during the perimenopausal period [54]. There has been a reported case of the spontaneous rupture of clear cell ovarian cancer, an uncommon type, in a 54-year-old postmenopausal patient. CT findings revealed a tumor with a diameter of approximately 18.2 cm, suggesting the possibility of malignancy, and a significant amount of hemoperitoneum was observed during surgery [55]. Secondly, most ovarian cancers present with nonspecific abdominal symptoms and are often diagnosed at an advanced stage, making acute abdomen rare [55]. Lastly, the pattern of the ovarian tumor observed through sonography and CT findings is crucial for differentiating between benign and malignant ovarian tumors [56]. To summarize, the serum tumor markers identified in this study should be used as additional findings in imaging tests, such as sonographic findings. It is important to note that this study is a retrospective analysis aimed at distinguishing between patients with ruptured and unruptured endometriosis.

There are several limitations to this study. Firstly, the study group was gathered between January 2013 and December 2014, which represents a slightly different approach to endometriosis treatment. When endometrioma was observed around 2013, the use of hormones before surgery was not recommended [57]. However, as of 2023, preoperative hormone therapy can be considered [12]. Therefore, this study has limitations in terms of its timing. Another limitation is the small sample size, as the incidence of ruptured endometrioma was very low.

It is important to emphasize the value of our study, as it represents the first report in the literature to undertake a comparison between patients with unruptured and ruptured endometrioma, especially aiming to identify preoperative diagnostic factors associated with the occurrence of ruptured endometrioma.

## 5. Conclusions

In conclusion, when confronted with a patient experiencing sudden, intense lower abdominal pain, an abdominal CT scan revealing an ovarian tumor accompanied by fluid echoes in both the pelvic cavity and extrapelvic regions, and elevated serum levels of CA125 (>100.9 IU/mL), CA19-9 (>27.7 IU/mL), and CRP (>1.0 g/dL), physicians should maintain a high level of suspicion for a ruptured endometrioma. In such cases, it is recommended that emergency surgical intervention be promptly carried out to achieve the most favorable prognosis possible (Figure 2).

## Figures and Tables

**Figure 1 jpm-13-01226-f001:**
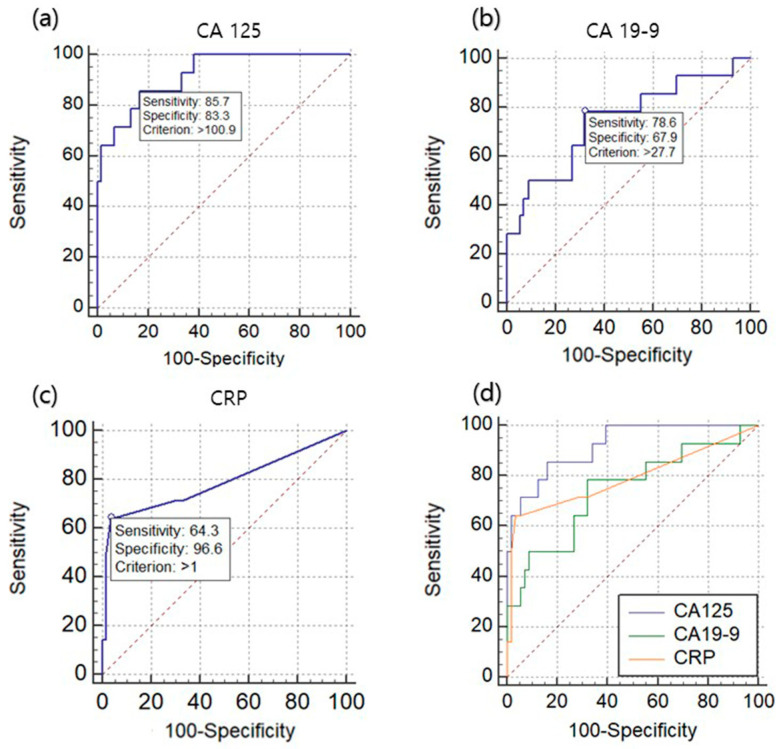
Cutoff value for diagnosing ruptured endometrioma. (**a**) Cutoff value of CA 125 for diagnosis of ruptured endometrioma; (**b**) cutoff value of CA 19-9 for diagnosis of ruptured endometrioma; (**c**) cutoff value of CRP for diagnosis of ruptured endometrioma; (**d**) cutoff value of CA 125, CA 19-9, and CRP for diagnosis of ruptured endometrioma.

**Figure 2 jpm-13-01226-f002:**
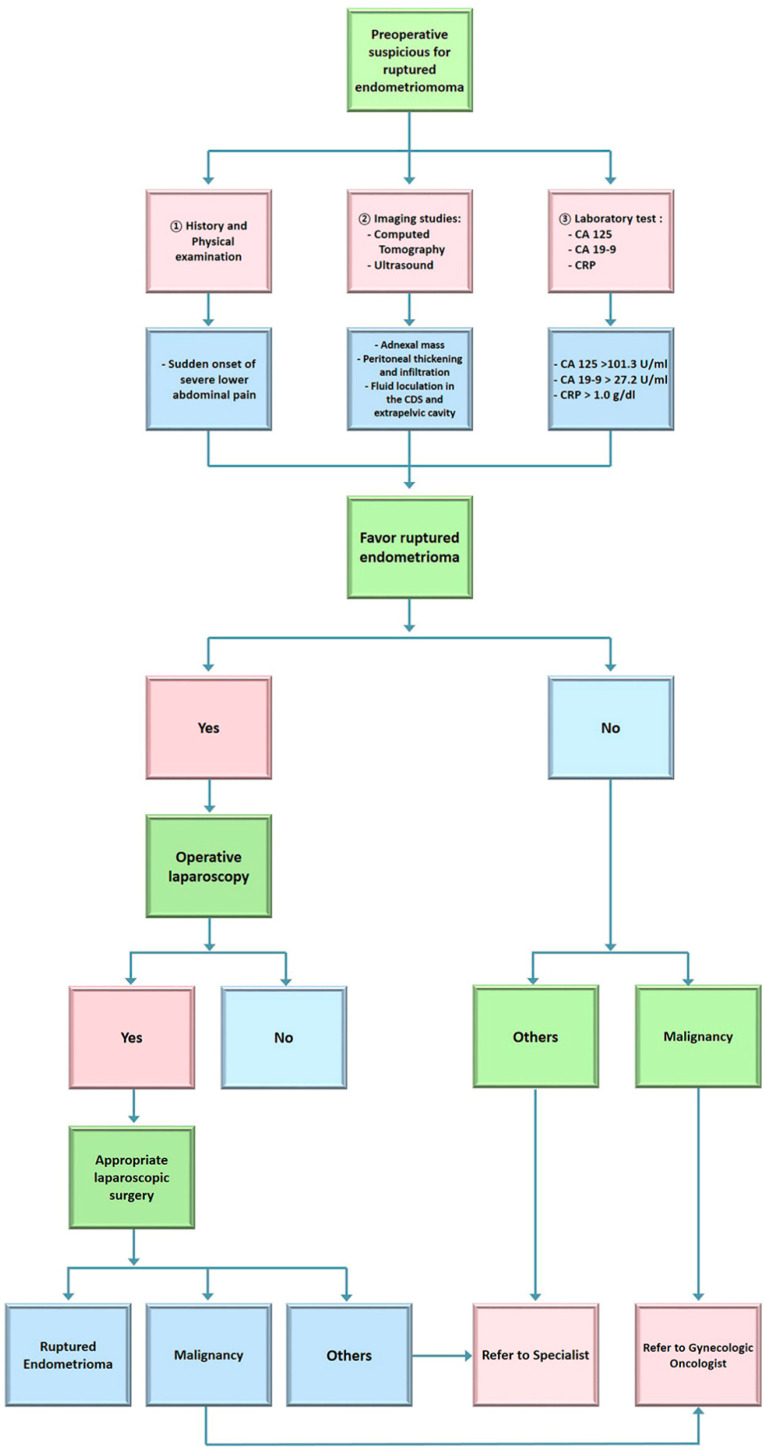
Flowchart for the diagnosis and treatment of suspicious ruptured endometrioma.

**Table 1 jpm-13-01226-t001:** General characteristics of patients with ruptured and unruptured endometrioma.

	Ruptured (14)	Unruptured (60)	*p*
Age (yrs)	30.57 (±1.932)	33.83 (±0.814)	0.169
Parity	0 (0–2)	0 (0–3)	0.138
BMI (kg/m^2^)	19.4 (17–23.4)	21.5 (17–32)	0.048
MCD (days)	13.5 (1–32)	18.0 (1–60)	0.1
Visit center			0.000
ER	14 (100)	8 (13.3)	
Outpatient clinics	0 (0)	52 (86.7)	
Chief complaint			0.000
Dysmenorrhea	0 (0)	13 (21.7)	
LAP	14 (100)	22 (36.7)	
Incidental	0 (0)	25 (41.7)	
Previous ES history	0	4 (6.67)	
CA 125 (U/mL)	345.1 (70.8–3363.0)	49.8 (9.2–317.4)	0.000
CA 19-9 (U/mL)	46.0 (1.0–1158)	19.1 (0.6–176.2)	0.005
CRP (mg/dL)	1.2 (0.3–16.6)	0.3 (0.3–12.0)	0.000

Note: BMI, body mass index; MCD, menstrual cycle day; ER, emergency room; LAP, lower abdominal pain; ES, endometriosis.

**Table 2 jpm-13-01226-t002:** Preoperative CT findings in patients with ruptured endometrioma.

Patients	Size of Mass (cm)	Fluid Collection					Perito-Neal Infiltration	Fluid Loculation	Preoperative Diagnosis
		CDS	RPCG	LPCG	Subhepatic	Lt. Subphrenic			
1	8	(+)	(+)	(−)	(+)	(−)	(+)	(+)	Ovarian cancer
2	6	(+)	(+)	(+)	(−)	(−)	(+)	(−)	Hemoperitoneum
3	7	(+)	(−)	(+)	(−)	(−)	(+)	(+)	Hemoperitoneum
4	5	(+)	(+)	(+)	(−)	(−)	(+)	(+)	Hemoperitoneum
5	7	(+)	(−)	(−)	(−)	(−)	(+)	(−)	Benign tumor
6	8	(+)	(+)	(+)	(+)	(−)	(+)	(+)	TOA
7	7	(+)	(−)	(−)	(−)	(−)	(+)	(+)	TOA
8	7	(+)	(+)	(−)	(−)	(−)	(+)	(+)	TOA
9	6	(+)	(−)	(−)	(−)	(−)	(+)	(+)	TOA
10	6	(+)	(+)	(−)	(−)	(−)	(+)	(−)	Endometriosis
11	8	(+)	(+)	(+)	(+)	(−)	(+)	(−)	Endometriosis
12	5	(+)	(−)	(−)	(−)	(−)	(+)	(−)	TOA
13	5	(+)	(−)	(−)	(−)	(−)	(+)	(−)	TOA
14	6	(+)	(+)	(+)	(−)	(+)	(+)	(+)	Hemoperitoneum
Numbers of Positive Findings		14	8	6	3	1	14	8	

Note: CDS, cul-de-sac; RPCG, right paracolic gutter; LPCG, left paracolic gutter; Lt, left; TOA, tubo-ovarian abscess; (+), positive finding; (−), negative finding.

## Data Availability

Data sharing is not applicable to this study because of privacy or ethical restrictions.

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
