# Peer review of "Diagnostic Clues for Women with Acute Surgical Abdomen Associated with Ruptured Endometrioma"

_jpm, 2023, doi:10.3390/jpm13081226_

Round 1

Reviewer 1 Report

The main focus of the paper is ruptured endometrioma, a benign disease for which the authors have investigated some tumoral markers (CA125 and CA19-9) and CRP to determine if the patient needs an emergency surgical intervention. 

CRP may be elevated in the simple setting of acute pain and minimal inflammation. 

The conclusion states that when confronted with an ovarian tumor and sudden onset abdominal pain in the setting of elevated CA125 and CA19-9 the patient needs emergency surgery. This may be misleading and should be revised. In the setting of an ovarian tumor, liquid in the peritoneal cavity and elevated tumoral markers, a ruptured endometrioma is the best case scenario but the worst case scenario is a ruptured malignant cystic ovarian tumor with peritoneal spillage. 

The authors should make this clearer and revise the manuscript regarding this differential diagnosis. This may be also a consequence of a retrospective study, a limitation recognized by the authors, because the authors now the final pathology by the time of the preparation of the manuscript. 

Author Response

Response to Reviewer 1 Comments

  1. The main focus of the paper is ruptured endometrioma, a benign disease for which the authors have investigated some tumoral markers (CA125 and CA19-9) and CRP to determine if the patient needs an emergency surgical intervention.CRP may be elevated in the simple setting of acute pain and minimal inflammation. The conclusion states that when confronted with an ovarian tumor and sudden onset abdominal pain in the setting of elevated CA125 and CA19-9 the patient needs emergency surgery. This may be misleading and should be revised. In the setting of an ovarian tumor, liquid in the peritoneal cavity and elevated tumoral markers, a ruptured endometrioma is the best case scenario but the worst case scenario is a ruptured malignant cystic ovarian tumor with peritoneal spillage.The authors should make this clearer and revise the manuscript regarding this differential diagnosis. This may be also a consequence of a retrospective study, a limitation recognized by the authors, because the authors now the final pathology by the time of the preparation of the manuscript.

Response 1: Thank you very much for this detailed suggestion. this comment. We absolutely agree with you, and the point you made is a very important thing. However, in general, acute abdomen from epithelial ovarian cancer (EOC) is very rare and ruptured tumors are known to occur more often in germ cell tumors or sex cord stromal tumors than in EOC. We have added these points into the discussion as follows: (page 8-9, lines 329-360)

================================================================

treatment of ruptured endometrioma to enhance the prognosis and reduce the risk of recurrence.

Additionally, it is important to consider the possibility of malignancy when an ovarian tumor is present and there is an elevation in serum tumor markers such as CA 125 or CA 19-9. Gynecologic causes of hemoperitoneum include ectopic pregnancy, hemorrhagic corpus luteal cyst, endometriosis, and uterine or ovarian neoplasms [48]. However, acute abdomen due to ovarian malignancy is rare [49]. Among these causes Among these causes, germ cell tumors and sex cord stromal tumors are the most common ovarian malignancies associated with spontaneous hemoperitoneum [50, 51]. Germ cell tumors are typically associated with serum tumor markers such as human chorionic gonadotropin (HCG) and alpha-fetoprotein (AFP) [52]. Sex cord-stromal tumors utilize various hormones such as estradiol, inhibin, testosterone, and androstenedione as tumor markers [53]. In cases of epithelial ovarian cancer, which is the most common malignancy associated with elevated CA 125 levels, it may be challenging to distinguish it from a rupture of an endometrioma. However, there are several important clues to differentiate between them. Firstly, epithelial ovarian cancer is rare in individuals under 40 years of age, with over 90% of cases occurring in those over 40 years old. Additionally, the incidence of epithelial ovarian cancer also increased with age, peaking at around 70 [54]. This study, being retrospective, only included women with confirmed biopsies of endometriomas, and the average age of the ruptured and unruptured groups was 30.575 years and 33.83 years, respectively, which is younger than the typical onset age for epithelial ovarian cancer. Furthermore, clear cell ovarian cancer, which is a rare histologic subtype, primarily develops during the perimenopausal period. [55]. There has been a reported case of spontaneous rupture of clear cell ovarian cancer, an uncommon type, in a 54-year-old postmenopausal patient. CT findings revealed a tumor with a diameter of approximately 18.2cm, suggesting the possibility of malignancy, and a significant amount of hemoperitoneum was observed during surgery [56]. Secondly, most ovarian cancers present with nonspecific abdominal symptoms and are often diagnosed at an advanced stage, making acute abdomen is rare [56]. Lastly, the pattern of the ovarian tumor observed through sonography and CT findings is crucial for differentiating between benign and malignant of ovarian tumor [57]. To summarize, the serum tumor markers identified in this study should be used as additional findings in imaging tests, such as sonographic findings. It is important to note that this study is a retrospective analysis aimed at distinguishing between patients with ruptured and unruptured endometriosis.

Reviewer 2 Report

The main question was clearly addressed by the research. The topic seems novel as i couldn't find a similar previous article had been published (in English).. It highlights the attention toward possible ruptured endometrioma in ie presenting with acute abdominal pain.

the methodology seems well constructed and patients with unruptured endometrioma were selected carefully as control

conclusion seems consistent with the evidence and arguments presented.

References are appropriate

tables and figure are clear and easy to read

Author Response

Response to Reviewer 2 Comments

The main question was clearly addressed by the research. The topic seems novel as i couldn't find a similar previous article had been published (in English). It highlights the attention toward possible ruptured endometrioma in ie presenting with acute abdominal pain.
the methodology seems well constructed and patients with unruptured endometrioma were selected carefully as control
conclusion seems consistent with the evidence and arguments presented.
References are appropriate
tables and figure are clear and easy to read

Response 2: I really appreciate everything you’ve done.

Reviewer 3 Report

The authors explored the pre-operative factors to identify the ruptured endometrioma comparing women with ruptured endometrioma and women without unruptured endometrioma. The idea of present study is interesting. Nevertheless, this study has several concerns to resolve. My comments to improve the manuscript are as follows.

This retrospective study was conducted between January 2013 and December 2014. Since this is a retrospective study, please clarify how the authors decided on the duration of the study. Moreover, the duration of present study is outdated. This point is a limitation of this study.

Do the authors have information about the treatment for endometriosis?

Table 1: Table legend is missing. Please also clarify the abbreviations.

Lines 157-158: Please revise this sentence according to my following comments.

Tests evaluated for screening have not detected ovarian cancer at an early enough stage to reduce mortality and have led to unnecessary surgical procedures for false-positive results. Studies show CA125 may predict ovarian cancer, but its usefulness is hampered by its limited specificity and very low positive predictive value.

The discussion is too long and please reduce the word count by 30%.

The small number of women with ruptured endometrioma is a limitation of this study.

Several cited studies were outdated. Please cite the latest studies.

While some sentences are difficult to read, the quality of English was acceptable.

Author Response

Response to Reviewer 3 Comments

The authors explored the pre-operative factors to identify the ruptured endometrioma comparing women with ruptured endometrioma and women without unruptured endometrioma. The idea of present study is interesting. Nevertheless, this study has several concerns to resolve. My comments to improve the manuscript are as follows.

  1. This retrospective study was conducted between January 2013 and December 2014. Since this is a retrospective study, please clarify how the authors decided on the duration of the study. Moreover, the duration of present study is outdated. This point is a limitation of this study.
  2. Do the authors have information about the treatment for endometriosis?

Response 3-1 and 3-2: We agree with the reviewer’s comment. I completely agree with your comment. The first time the PI saw a patient with a ruptured endometrioma was in 2013, and the first time we designed this study and collected data was in 2015, as we were seeing similar patients frequently. In 2016, after cleaning the data and having only statistical results, the study was stopped due to personal reasons. In the meantime, trends in the treatment of endometriosis have changed. However, ruptured endometrioma is a condition that requires surgery, and we have experienced good results when applying our findings, so we decided to publish them in a paper, albeit a long time ago. We described this limitation in the Discussion session (Page 9, line 365-370)

================================================================

such as sonographic findings. It is important to note that this study is a retrospective analysis aimed at distinguishing between patients with ruptured and unruptured endometriosis.

There are several limitations to this study. Firstly, the study group was gathered between January 2013 and December 2014, which represents a slightly different approach to endometriosis treatment. When endometrioma was observed around 2013, the use of hormones before surgery was not recommended [58]. However, as of 2023, preoperative hormone therapy can be considered [59]. Therefore, this study has limitations in terms of its timing. Additionally, the small sample sized is also a limitation.

  1. Table 1: Table legend is missing. Please also clarify the abbreviations.

Response 3-3: Thank you for your detailed suggestion. We added the abbreviations below the table.

Note: BMI, Body mass index; MCD, Menstrual cycle day; ER, Emergency room; LAP, Lower abdominal pain; ES, Endometriosis

  1. Lines 157-158: Please revise this sentence according to my following comments.

Response 3-4: Thank you for your kind comment. We revise the sentence according to your comments (line 160-163)

================================================================

CA125 is a mucinous glycoprotein with a high molecular weight, derived from coelomic epithelia found in various locations such as the endometrium, fallopian tube, ovary, and peritoneum [8, 9]. Tests evaluated for screening have not detected ovarian cancer at an early enough stage to reduce mortality and have led to unnecessary surgical procedures for false-positive results. Studies show CA125 may predict ovarian cancer, but its usefulness is hampered by its limited specificity and very low positive predictive value. Furthermore, it is important to note that elevated levels of CA125 can be observed not only in cases of ovarian cancer but also due to several benign causes, and abnormal levels can also be attributed to non-gynecologic factors.

  1. The discussion is too long and please reduce the word count by 30%.

Response 3-5: I totally agree with you too. We realized that the discussion is too long. However, JPM editorial office requires a minimum of 4,000 word counts. So, we have added the contents of the discussion as well.

  1. The small number of women with ruptured endometrioma is a limitation of this study.

Response 3-6: Thank you for the sharp comment. We know the limitation of the small number of patients with ruptured endometrioma. However, the incidence of ruptured endometrioma was too small. The percentage of surgery for endometriosis was 11.5% of all gynecologic surgeries [1]. Among these, 3.4% undergo surgery due to ruptured endometrioma [2]. We described this limitation in the Discussion session (Page 9, line 370)

================================================================

There are several limitations to this study. Firstly, the study group was gathered between January 2013 and December 2014, which represents a slightly different approach to endometriosis treatment. When endometrioma was observed around 2013, the use of hormones before surgery was not recommended [58]. However, as of 2023, preoperative hormone therapy can be considered [59]. Therefore, this study has limitations in terms of its timing. Additionally, the small sample sized is also a limitation.

References of response 3-6)

  1. McLeod BS, Retzloff MG. Epidemiology of endometriosis: an assessment of risk factors. Clin Obstet Gynecol 2010; 53:389-96.
  2. Kataoka T, Watanabe Y, Hoshiai H. Retrospective evaluation of tumor markers in ovarian mature cystic teratoma and ovarian endometrioma. J Obstet Gynaecol Res 2012; 38:1071-6.

  1. Several cited studies were outdated. Please cite the latest studies.

Response 3-7: We agree with the reviewer’s comment. We changed the reference to the latest studies. Renumbered from No. 52 due to the addition of reference No. 51 (From 52 to 53, from 53 to 54, from 55 to 56, from 56 to 57). And new reference was added as No. 58. The list is as follows:

  1. Cruickshank, D. Screening for ovarian cancer by CA-125 measurement. The Lancet 1988, 331 (8584), 540-541.

à 9. Høgdall, E. V.; Christensen, L.; Kjaer, S. K.; Blaakaer, J.; Kjærbye-Thygesen, A.; Gayther, S.; Jacobs, I. J.; Høgdall, C. K. CA125 expression pattern, prognosis and correlation with serum CA125

  1. Chen, F.-P.; Soong, Y.-K.; Lee, N.; Kai Lo, S. The use of serum CA-125 as a marker for endometriosis in patients with dysmenorrhea for monitoring therapy and for recurrence of endometriosis. Acta obstetricia et gynecologica Scandinavica 1998, 77 (6), 665-665.

à 15. Coutinho, L. M.; Ferreira, M. C.; Rocha, A. L. L.; Carneiro, M. M.; Reis, F. M. New biomarkers in endometriosis. Advances in Clinical Chemistry 2019, 89, 59-77.

  1. Ye, C.; Ito, K.; Komatsu, Y.; Takagi, H. Extremely high levels of CA19-9 and CA125 antigen in benign mucinous ovarian cystadenoma. Gynecologic oncology 1994, 52 (2), 267-271.

à 27. Sagi-Dain, L.; Lavie, O.; Auslander, R.; Sagi, S. CA 19-9 in evaluation of adnexal mass: retrospective cohort analysis and review of the literature. The International journal of biological markers 2015, 30 (3), 333-340.

  1. Watanabe, J.; Johboh, T.; Hata, H.; Kuramoto, H. Clinical significance of CA19-9 for endometriosis. Nihon Sanka Fujinka Gakkai Zasshi 1990, 42 (2), 155-161.

à 29. Pandey, D.; Wasinghon, P.; Huang, K.-G. Laparoscopic Management of Peritonitis Due to a Ruptured Ovarian Endometrioma with Extremely High Levels of Cancer Antigen–125 and Cancer Antigen–19-9. Journal of Gynecologic Surgery 2019, 35 (3), 198-200.

  1. Roberts, W. L.; Moulton, L.; Law, T. C.; Farrow, G.; Cooper-Anderson, M.; Savory, J.; Rifai, N. Evaluation of nine automated high-sensitivity C-reactive protein methods: implications for clinical and epidemiological applications. Part 2. Clinical chemistry 2001, 47 (3), 418-425.

à 38. Pathak, A.; Agrawal, A. Evolution of C-reactive protein. Frontiers in immunology 2019, 10, 943.

  1. Umaria, N.; Olliff, J. Imaging features of pelvic endometriosis. The British Journal of Radiology 2001,74 (882), 556-562.

à 42. Coutureau, J.; Mandoul, C.; Verheyden, C.; Millet, I.; Taourel, P. Acute abdominal pain in women of reproductive age: keys to suggest a complication of endometriosis. Insights into Imaging 2023, 14 (1), 94.

  1. Nelson, B. E.; Carcangiu, M. L.; Chambers, J. T. Intraabdominal hemorrhage with pulmonary large cell carcinoma metastatic t to the ovary. Gynecologic oncology 1992, 47 (3), 377-381.

à 49. Stewart, C.; Ralyea, C.; Lockwood, S. Ovarian cancer: an integrated review. In Seminars in oncology nursing, 2019; Elsevier: Vol. 35, pp 151-156.

  1. Geisler, J. P.; Denman, B. J.; Cudahy, T. J.; Lee, T. H.; Geisler, H. E. Ovarian carcinoma presenting as intra-abdominal hemorrhage. Gynecologic oncology 1994, 53 (3), 380-381.

à 50. Schultz, K. A. P.; Harris, A. K.; Schneider, D. T.; Young, R. H.; Brown, J.; Gershenson, D. M.; Dehner, L. P.; Hill, D. A.; Messinger, Y. H.; Frazier, A. L. Ovarian sex cord-stromal tumors. Journal of oncology practice 2016, 12 (10), 940-946.

  1. Göbel, U.; Schneider, D.; Calaminus, G.; Haas, R.; Schmidt, P.; Harms, D.; MAKEI, G. Germ-cell tumors in childhood and adolescence. Annals of oncology 2000, 11 (3), 263-272.

à 52. Low, J. J.; Ilancheran, A.; Ng, J. S. Malignant ovarian germ-cell tumours. Best practice & research Clinical obstetrics & gynaecology 2012, 26 (3), 347-355.

Round 2

Reviewer 1 Report

The authors have made significant improvements regarding the limitations and biases of the study. 

Reviewer 3 Report

The authors have revised the manuscript appropriately.

This manuscript was easy to read.